# A Review on Heat Transfer of Nanofluids by Applied Electric Field or Magnetic Field

**DOI:** 10.3390/nano10122386

**Published:** 2020-11-29

**Authors:** Guannan Wang, Zhen Zhang, Ruijin Wang, Zefei Zhu

**Affiliations:** School of Mechanical Engineering, Hangzhou Dianzi University, Hangzou 310000, China; wgn191010006@hdu.edu.cn (G.W.); zhangzhen1997@hdu.edu.cn (Z.Z.)

**Keywords:** nanofluids, heat transfer enhancement, applied electric field, applied magnetic field, thermal conductivity

## Abstract

Nanofluids are considered to be a next-generation heat transfer medium due to their excellent thermal performance. To investigate the effect of electric fields and magnetic fields on heat transfer of nanofluids, this paper analyzes the mechanism of thermal conductivity enhancement of nanofluids, the chaotic convection and the heat transfer enhancement of nanofluids in the presence of an applied electric field or magnetic field through the method of literature review. The studies we searched showed that applied electric field and magnetic field can significantly affect the heat transfer performance of nanofluids, although there are still many different opinions about the effect and mechanism of heat transfer. In a word, this review is supposed to be useful for the researchers who want to understand the research state of heat transfer of nanofluids.

## 1. Introduction

According to the second law of thermodynamics, heat will be transferred from high temperature to low temperature due to the temperature gradient. This energy transfer caused by temperature difference is termed “heat transfer”. The heat transfer process of objects can be divided into three basic heat transfer modes: heat conduction, heat convection and heat radiation. Convection heat transfer of working fluid is widely used in mechanical engineering, energy engineering, chemical engineering. For instance, the successful heat exchange process and the stable control of the temperature in a nuclear reactor are crucial. Hence, we have been looking for ways to enhance heat transfer. The approaches of heat transfer enhancement generally include active and passive approaches. The active heat transfer enhancement includes mechanical stirring, surface vibration, fluid pulsation, applied electric or magnetic field, while special-shaped tube and internal plug-in, splitting and combining are belong to the passive heat transfer enhancement. However, both approaches do not change the nature of the working fluid but produce chaotic convection. In fact, one of the fatal defects of thermal engineering devices is the low thermal conductivity of conventional fluids such as water, ethylene glycol or oil, which results in the lower heat transfer efficiency in such devices. Nevertheless, nanofluids with higher thermal conductivity can remove such a defect successfully [1]. Nanofluid is a new type of uniform, stable, and high thermal conductivity heat transfer medium formed by dispersing metal or nonmetal nanoparticles into traditional media such as water, alcohol, and oil. Choi et. al [2] first proposed the concept of nanofluids and preliminarily investigated the addition of nanosized particles to the fluid in 1995. The results showed that the thermal conductivity of the fluid was significantly improved. Since then, a large number of research on heat transfer enhancement by nanofluids have been investigated.

Eastman et al. [3] compared the thermal conductivity between pure ethylene glycol and Cu-ethylene glycol nanofluid. Then they found that the Cu-ethylene glycol nanofluid containing 0.3 vol % Cu nanoparticles with diameter less than 10 nm can increase the effective thermal conductivity by up to 40% contrasted with pure ethylene glycol.

Yousefi et al. [4] examined the heat transfer performance of the flat-plate collector with pure and Al_2_O_3_-water nanofluid. Outcomes showed that the Al_2_O_3_-water nanofluid could improve heat transfer efficiency by up to 28.3% compared with pure water. Das et al. [5] also found that adding alumina nanoparticles into the water can significantly improve the thermal conductivity of the working fluid, and the increase of temperature and concentration will improve the thermal conductivity.

Rao et al. [6,7] carried out a study on improving the cooling performance of the battery thermal management system (BTM) by Al_2_O_3_–water and Cu-water nanofluids, respectively, and compared the water-based BTM with the nanofluid based BTM. The results showed that the nanofluid based BTM could effectively improve cooling performance and decrease the average temperature of the battery.

In addition, nanofluids can also be used in food processing. Jafari et al. [8] investigated the possibility of using Al_2_O_3_–water nanofluids to replace the traditional pasteurization process for heat treatment, to shorten the time of heat treatment and improve the nutrition of fruit juice. In order to find the best nanofluid concentration, temperature and heat treatment time, three kinds of nanofluids with different concentrations were tested at different temperatures, and the comparative study was carried out under three different heat treatment durations of 30 s, 60 s and 90 s. Their work showed that higher concentrations of nanoparticles and lower temperatures or times result in higher vitamin C contents. These findings demonstrated for the first time that nanofluid heat transfer technology could be applied to improve the nutritional contents of food. Kah et al. [9] thought nanotechnology also could be used in agriculture, which may be helpful for enhancing crop nutrition and addressing inefficiencies in agriculture such as the inefficient use of water, fertilizers and energy.

Moreover, Wang et al. [10,11,12] explored the application of the Al_2_O_3_–water nanofluid to a microchannel heat sink (MCHS), and the results depicted that the application of nanofluid can improve the thermal efficiency of MCHS, thus ensuring the working temperature and photoelectric conversion efficiency of solar cells. Nanofluids are also used in cancer treatment [13,14,15], wastewater decontamination [16], solar distillation system [17], engine cooling system [18], micro high strength refrigeration system [19].

Nowadays, nanofluids are considered to be the next-generation heat transfer fluids because they offer new possibilities for increasing thermal conductivity. Compared with conventional heat transfer fluids and fluids containing microsized metallic particles, they have better thermal performance. Otherwise, nanoparticles have a much larger relative surface area compared with conventional particles, which cannot only significantly improve the heat transfer capabilities but also improve the stability of suspensions [20].

After nanofluids gradually became the mainstream heat transfer working fluid, researchers began to investigate how to improve the heat transfer performance of nanofluids. Common methods include changing the shape of nanoparticles [21], heat radiation [22,23,24,25,26,27], applied electric field [28,29], applied magnetic field [30] and applied electromagnetic field [31]. The heat transfer enhancement by an applied electric or magnetic field is because the electric force or magnetic force on nanoparticles can change the microstructure of nanofluid and movement of nanoparticles [32]. In addition, the interplay among the magnetic field, electric field, flow field and temperature field can result in more severe chaotic convection. Electric field enhanced heat transfer is furnished with many advantages such as ease of control, low power requirements, and simple design [33]. Since Chubb first studied the effect of electric field on heat transfer enhancement at the beginning of this century, many researchers have begun to investigate the effect of electric field on heat transfer enhancement. Jones et al. [34], Allen et al. [35] and Laohalertdecha et al. [36] successively reviewed the papers on electric field enhanced heat transfer since the last century. However, most of the previous studies only focused on the enhanced heat transfer of single-phase fluid by an electric field and magnetic field. In recent years, the research on heat transfer enhancement of nanofluids by applied electric and magnetic fields has attracted a large number of researchers’ attention, but they didn’t agree on the effect and mechanism, which limits the further study of heat transfer of nanofluids. There is still no definite conclusion on the thermal conductivity mechanism of nanofluids because so many possible mechanisms are proposed in the recent two three decades. Keblinski et al. [37] and Eastman et al. [38] proposed four possible mechanisms such as Brownian motion of the nanoparticles, molecular-level layering of the liquid at the liquid/particle interface, the nature of heat transport in the nanoparticles, and the effects of nanoparticle clustering [20], while there is another mechanism in solids at nanoscales such as the interface interaction between nanoparticles and the matrix that can impede the movement of phonons thus hindering heat transfer [39]. It is confirmed that Brownian motion and micro convection of nanoparticles contribute greatly to the thermal conductivity of nanofluids when the volume fraction of nanoparticles being lower. The model of nanofluid thermal conductivity proposed by Xuan et al. [40] contains the effect of Brownian movement. From the Xuan model, the nanoparticle enhances the thermal conductivity includes two aspects: the first is that the nanoparticles change the fluid composition and lead to the base fluid become a suspension, subsequently affecting the energy transport process. The second aspect is that the Brownian motion of the nanoparticles, as well as the interfacial interactions between the particles and the liquid molecules, intensify the energy transport [41]. Reversely, Evans et al. [42] indicated the effect of Brownian motion on thermal conductivity can be neglected.

It is proved by molecular dynamics (MD) simulations, the nanolayer at the interface of nanoparticle and base liquid can elevate the thermal conductivity of nanofluid due to its ordered structure [43,44]. However, it can be concluded from the theory of phonon heat transfer that the interfacial nanolayer may be unbeneficial to elevate the thermal conductivity of nanofluids owing to the increase of phonon scattering [45].

Keblinski et al. [37] indicated that the aggregation of nanoparticles could build a low thermal resistance channel, which can reduce the loss of heat transfer, thus increasing the heat transfer efficiency of nanofluids. In addition, Wang et al. [46] found the aggregation of nanoparticles can affect the evaporation of nanofluids and proposed the D^h^ theorem, which indicates that the evaporation time of the liquid is proportional to the h power of the initial diameter of the droplet. Besides, h is related to the volume fraction and distribution of nanoparticles. With the increase of the volume fraction of lyophobic nanoparticles, h will be greater, while h will be smaller with the increase of the volume fraction of lyophilic nanoparticles. On the contrary, Evans et al. [47] et al. indicated that the aggregation of nanoparticles is inessential on the thermal conductivity because the thermal conductivity is related to the aggregate morphology of nanoparticles [48]. Finally, Nagvenkar et al. [49] summarized that Brownian motion might be the main mechanism of heat conduction enhancement at a low particle concentration of nanoparticles, while the aggregation of nanoparticles is the main mechanism at high particle concentration.

Moreover, the thermophoresis force generated by temperature gradient also has a great influence on the movement of nanoparticles. Buongiorno et al. [50] indicated that temperature gradient and thermophoresis could cause significant changes in the properties of nanofluids within the flow boundary layer, in which the thermophoresis velocity is affected by the size of nanoparticles. The thermophoresis forces could consequently be utilized to sort micro/nanoparticles [51,52].

Recently, Tzou et al. [53] and the team of Wang Liqiu et al. [54,55] used a two-phase hysteresis model to investigate the heat transfer of nanofluids. By analyzing and measuring the heat flux vector and time lag ratio of different nanofluids, the existing conditions of the thermal wave were obtained, and the thermal wave theory of heat transfer of nanofluids was established. Contrary to the practice of Tzou et al. [53], Dolatabadi et al. [56] tried to clarify the energy transfer mechanism at the micro/nanoscale by analyzing the structure of the solid–liquid interface and established a model of macroscopic thermal conductivity. However, the inner mechanism of heat conduction of nanofluids is not clear, the temperature is discontinuous in the microscale heat transfer, and the heat flow vector and temperature gradient are separated. Moreover, there is the non-Fourier conduction of heat transfer of nanofluids, which needs further study.

The interfacial thermal resistance caused by phonon scattering is the only mechanism to reduce the thermal conductivity of nanofluids, and the solid–liquid interface structure is very important for the thermal conductivity of nanofluids. When the nanoparticles are added into the liquid, the dehydrated counter ions strongly combine with the surface charged groups to form a fixed ordered structure, which is a nanolayer with a thickness of about 1 nm [57]. Otherwise, the ion binding in the external region (diffusion layer) is weakened, resulting in a gradual decrease in electric potential. The thickness of the diffusion layer is generally several to hundreds of nanometers, and it is related to the size of nanoparticles, surface potential, ion concentration and pH value [58].

Lee et al. [59] and Jung et al. [60] indicated that the surface charge state and the electric double layer are the main factors for the increase of thermal conductivity of nanofluids because the surface charge state determines the structure of ions adsorbed on the surface of nanoparticles, the resistance of heat and phonon transfer channels. Besides, the amount of charge determines the number of transfer channels. The structure of the electric double layer and the thermal conductivity is different when the nanoparticles and the base liquid are different [61]. These are confirmed by many published results. For example, Ha et al. [62] compared the effects of the interface layer, aggregate and electric double layer on the effective thermal conductivity of TiO_2_-water nanofluids, and then they found that the electric double layer had the greatest influence. Zhao et al. [63] indicated that the electric double layer has an important influence on the flow and heat transfer in the microchannel by using homotopy analysis methods. From the molecular dynamics results, Cui et al. [64] found that the specific surface area and the electric double layer should be taken into account when the particle size affects the thermal conductivity. Mitiche et al. [65] investigated the effect of the interface layers in enhancing the heat transfer performance of Cu-Ar nanofluids with the linear response theory and equilibrium molecular dynamics simulations. They found that the thermal conductivity of the nanofluid is related to the vibration state of the Ar atoms around the nanoparticles, and the vibration frequency of Cu atoms of the nanoparticles is consistent with liquid Ar atoms around Cu nanoparticles in a large frequency domain.

The flow and heat transfer of nanofluids in micro/nanochannels under an applied electric field have attracted extensive attention. For example, the effect of wall potential on electroosmotic velocity [66], the analysis of four kinds of electrohydrodynamic phenomena as electroosmosis, electrophoresis, flow potential and deposition potential [67], and the influence of heat transfer of nanofluids in micro/nanochannels [68]. In addition, some researchers used acoustic waves [69] to control the flow and heat transfer in micro/nanochannels, which provides an effective way to improve the heat transfer by controlling the electric double layer structure in the applied field.

Therefore, the purpose of this paper is to explore the effect of the applied electric field or magnetic field on the heat transfer performance and mechanism of nanofluids in the past decade. In addition, the interfering factors as the Nusselt, Hartmann, Reynolds, Darcy, Rayleigh, and Prandtl numbers can affect the heat transfer of nanofluids in the existence of the electric field or magnetic field are also analyzed. The Nusselt number (*Nu*) is a dimensionless number for the intensity of convective heat transfer, and a larger Nusselt number indicates more active convection. Hartman number (*Ha*) is the ratio between electromagnetic force and viscous force, which increases with the enhancement of the magnetic field. The Reynolds number is a physical representation of the ratio of inertial forces to viscous forces. When the Reynolds number is small, the influence of viscous force is dominant, and when the Reynolds number increases, the influence of inertia becomes more significant. Darcy number (*Da*) represents the permeability of porous media. The higher the value of Darcy number, the greater the permeability of porous media. Rayleigh number (*Ra*) is a dimensionless number related to natural convection. When the Rayleigh number of the fluid is lower than the critical value, the main form of heat transfer is heat conduction, and when the Rayleigh number exceeds the critical value, the main form of heat transfer is heat convection. The Prandtl number represents the relative size of the thickness of the flow boundary layer and the thermal boundary layer. When the Prandtl number increases, the thickness of the flow boundary layer increases relative to the thickness of the thermal boundary layer.

We searched the literature on Web of Science, and the keywords are “nanofluid”, “electric field”, “heat transfer”. The flowchart of this survey methodology is shown in Figure 1.

## 2. Effect of Electric Field on Heat Transfer

### 2.1. The Mechanism of Heat Transfer Enhancement by Electric Field

Through experimental research and theoretical demonstration, researchers found that the electric field has a great influence on the heat conduction of single-phase and two-phase fluids. For the single-phase liquid, the mechanism of electric field enhancing heat transfer is mainly due to the pressure difference caused by the uneven distribution of electric field and the spatial variation of dielectric constant, which results in the electric convection inside the fluid and increases the disturbance degree of the fluid. As for the two-phase liquid, the external field also produces electric convection in the flow field, which makes the fluid move more violently and thus enhances the heat transfer of the fluid. Based on the model proposed by Xuan et al. [40], Chen et al. [41] proposed the mechanism of enhancing heat transfer of nanofluids by an electric field. They indicated that when the fluid is subjected to the electric field, the nanoparticles in the base fluid are affected by the electrophoretic force and dielectric electrophoretic force due to the existence of an electric double layer near the nanoparticles and zeta potential on the surface of nanoparticles, while Zeta potential is the potential difference between the continuous phase and the stable fluid layer attached to the dispersed particles. These additional forces can enhance the motion and range of movement of nanoparticles in the base liquid. Thus, nanoparticles can transfer energy in a wider range. While Liu et al. [70] proposed another theory, in which the enhancement of heat transfer of nanofluids in the presence of electric field is due to the fact that electrophoretic force, dielectric electrophoretic force and electrostrictive force can enhance the movement of nanoparticles, increase the chance of collision between particles, help to destroy the flow boundary layer, and thus strengthen the heat transfer between nanoparticles. Moreover, they also indicated that the enhancement of heat transfer of nanofluids with the applied electric field is due to the acceleration of the movement of nanoparticles by electric field, because the electric field can provide the external force, so the fast movement speed of nanoparticles increases the thermal conductivity of nanofluids. Moreover, the acceleration of the movement of nanoparticles by an electric field is affected by temperature because the increase of temperature can reduce the viscosity of nanofluids, and the acceleration effect is better when viscosity is low.

Moreover, the electroviscous effect is the main mechanism affecting the flow and heat transfer of nanofluids. Rubio-Hernändez et al. [71] found that the primary electric adhesion is caused by the deformation of the double electric layer, the secondary electric adhesion is caused by the overlap of the double electric layer, and the third-order electric adhesion is caused by the change of particle shape and size. Electric adhesion is an electrostatic effect in which two surfaces shrink in the existence of an electric field. It can be seen that the double electric layer plays an important role as a mechanism for the momentum transfer of nanofluids. Unfortunately, there are few reports on the mechanism of energy transfer between the electric double layers of charged nanoparticles. Based on the basic theory of molecular thermal vibration, Kang and Wang [72] explored the influence of electric field coupling of charged nanoparticles on the vibration response of nanoparticles and obtained the calculation method of direct energy transfer between nanoparticles, but they did not consider the electric field change in diffusion layer and the effect of the applied electric field mentioned in reference [73]. Due to the direct energy exchange between nanoparticles, it is possible to generate heat flow before the temperature gradient produces heat flow, resulting in non-Fourier conduction. When measuring the thermal conductivity of nanofluids by the transient hot wire method, the experimental value is often higher than the predicted value, which is due to the existence of heatwave heat transfer [74].

### 2.2. Effect of Electric Field on Heat Transfer Enhancement

At the beginning of electrohydrodynamics (EHD) development in the last century, researchers mainly focused on the heat transfer enhancement of single-phase fluid by an electric field. However, researchers gradually began to focus on the heat transfer enhancement of nanofluid by an electric field in the new century because of the increasing research works and applications of nanofluid in heat transfer. In recent decades, many researchers have explored the influencing factors, clarified the heat transfer laws of nanofluids with an applied electric field by establishing theoretical models, numerical simulation and experimental research, and finally found that the heat transfer and convection of nanofluids increase with the enhancement of electric field strength. The factors that can affect the enhancement effect of electric fields on nanofluid heat transfer are shown in Table 1.

The study of Daniel et al. [97], Zhao et al. [98] and Tang et al. [99] have confirmed the heat transfer law of nanofluid in the presence of an applied electric field, that is, with the increase of electric field strength, the heat transfer performance of nanofluids improves. Daniel et al. [97] examined the EHD flow of nanofluids toward a non-linear tensile surface with the applied electric field and explored the heat transfer characteristics with thickening phenomenon and applied electric field. Outcomes demonstrated that the electric field increases the flow rate and temperature of nanofluids. Zhao et al. [98] examined the effect of an electric field on the transportation of Fe_3_O_4_ nanoparticles through a covered porous cavity. They obtained results such as the curves of voltage, radiation and shape of nanoparticles by control volume finite element method (CVFEM). The results showed that the Nusselt number would increase with the increase of permeability, and the electric field can enhance the convection of nanofluid. Tang et al. [99] utilized CVFEM to investigate the computational simulation of EHD flow of nanomaterials. Results demonstrated that both the flow and Nusselt number of nanofluids are enhanced by applying an electric field, and the strengthening effect increases with the applied voltage. In addition, the application of higher voltage will make the isotherm near the tank wall more complex and generate a thermal plume. Figure 2a,b shows the contours for isotherm of the applied electric field voltage of 0 kV and 10 kV, respectively, when other parameters such as Reynolds number, Darcy number, radiation parameter and heat flux are fixed.

One of the reasons why the applied electric field can enhance heat transfer of nanofluids is that the electric field exerts three external forces on the nanofluids as electrophoretic force, dielectrophoretic force and electrostriction force. Due to the charges of nanoparticles, the nanoparticles will be affected by electrophoretic force in the existence of an electric field, while the generation of dielectrophoretic force is due to the gradient of electric field and the difference of dielectric constant between nanoparticles and base fluid. Besides, high inhomogeneity of electric field distribution will lead to the generation of electrostriction force. Although the mechanisms described by Liu et al. [70] and Chen et al. [41] in the previous section are different, one thing is the same that nanofluids are subjected to the external force imposed by the electric field. Thus the movement of nanoparticles is strengthened, and finally, the heat transfer of nanofluids is enhanced. Therefore, electrophoretic force, dielectrophoretic force and electrostriction force are vital to studying the heat transfer law of nanofluids enhanced through the application of the electric field. Kunti et al. [100] considered the interaction of electrophoretic force, dielectrophoretic force and electrostriction force in the scaling analysis of EHD convection transport. They found that the strength of the electric field force mainly depends on electrophoretic force and electrostriction force. The electrostriction force changes with the square of the gradient of dielectric constant and electric potential, while the electrophoretic force changes with the potential difference between the two electrodes [100]. Moreover, they also found that the electrophoretic force and the electrostriction force are proportional to the viscous force, and there is a competition between the two forces. However, they both can enhance the heat transfer of nanofluids, no matter which force is dominant. When the dominant forces in the nanofluid are different, the electric Rayleigh numbers (i.e., potential) in proportion to them are also different. Therefore, it is obvious that electrophoretic force and electrostriction force play a dominant role in the external force of the applied electric field on nanofluids. The contribution of the dielectrophoretic force is, instead, very small. Liu et al. [70] pointed out that whether the electric field is uniform or not has a great influence on the force generated by the electric field. The nanofluid under a uniform electric field is mainly affected by the electrophoretic force, while the nanofluid under a non-uniform electric field is mainly affected by the electrostriction force. This is different from the single-phase flow.

Wang et al. [101] pointed out that the single-phase fluid is mainly affected by the electrostrictive force in the non-uniform electric field, while the electrophoretic force and dielectric electrophoretic force are negligible. Besides, in the uniform electric field, the electrophoretic force and dielectric electrophoretic force are very small, and there is no electrostrictive force, so the electric field has almost no strengthening effect. In addition, they also explored the effect of gravity on heat transfer enhanced by applying an electric field. Outcomes demonstrated that the effect of electric field on heat transfer is obvious when there is no gravity, and the effect increases with the enhancement of voltage and heat flux. For the coupled convection caused by gravity and electric field force, the electric field force will destroy the natural convection caused by gravity, so the application of the electric field at low voltage will reduce the heat transfer performance. Although the heat transfer gradually increases with the enhancement of voltage, the effect is limited. Bao et al. [68] also examined the effect of gravity on electric field enhanced heat transfer. They found that EHD produces electric convection in the absence of gravity, and the heat transfer enhancement effect is remarkable. Moreover, the EHD effect is enhanced with the increase of voltage, while in the existence of gravity, the electric convection generated by an electric field and natural convection caused by gravity can counteract each other, so the EHD effect is weakened. Saravani et al. [75] investigated the effect of pressure and electric field on the convective heat transfer of nanofluids. The results showed that when the electric field is constant, and the pressure is increased, the Nusselt number decreases. However, the Nusselt number increases with the increase of the electric field at a fixed pressure. From the above analysis, it is found that the pressure is negatively correlated with the heat transfer of nanofluids, while the electric field is positively correlated.

In addition, there are many other factors that can affect the enhancement effect of electric field on nanofluid heat transfer, such as Reynolds number, Rayleigh number, Darcy number and heat radiation. Sheikholeslami et al. [76,77,78,79,80,81,82,83,84,85,86,87,88,89,90,91,92] and Safarnia et al. [93] investigated the effect of the applied electric field on the flow and heat transfer of Fe_3_O_4_-ethylene glycol nanofluid and Fe_3_O_4_-H_2_O nanofluid. Based on the control volume-based finite element method (CVFEM), various parameters such as Reynolds number, Rayleigh number, the volume fraction of nanoparticles, supplied voltage, radiation parameter, and Darcy number were numerically simulated. The results showed that the applied voltage has a great influence on the shape of the flow field, and with the application of Coulomb force, the distribution of isotherms near the wall will become denser. Besides, the distortion of isotherm increases because of the rise of Darcy number, radiation parameters and Coulomb force. The heat radiation can increase the temperature gradient of the nanofluid and thus improves the heat transfer performance of the nanofluid. In addition, the thermal conductivity of nanofluids increases with the enhancement of supplied voltage, Rayleigh number and Reynolds number, in which the effect of electric field on heat transfer is more significant at low Reynolds number and low Rayleigh number. Saleem et al. [94] and Truong Khang et al. [95] also investigated the influence of external electric field on the flow and heat transfer of Fe_3_O_4_-ethylene glycol nanofluid, and finally had a similar conclusion with Sheikholeslami et al. [76,77,78,79,80,81,82,83,84,85,86,87,88,89,90,91,92].

The research of Lu et al. [96] can be regarded as a supplement to the research of Sheikholeslami et al. [76,77,78,79,80,81,82,83,84,85,86,87,88,89,90,91,92]. They numerically investigated the electrothermal hydrodynamic flow of dielectric liquid in the presence of Coulomb force and buoyancy force and used a high-resolution upwind scheme to explore the heat transfer. It was found that the electric field can enhance the heat transfer more effectively for the fluid with a larger Prandtl number at a lower Rayleigh number. Moreover, they added that lower mobility parameters are also conducive to enhanced heat transfer in electric fields.

The study of Asadzadeh et al. [33] also supports the conclusion of Sheikholeslami. They studied the natural convection heat transfer of Fe_3_O_4_-ethylene glycol nanofluid on a thin platinum wire with an applied electric field. The experimental results showed that the electric field could enhance the natural convection heat transfer of ethylene glycol and Fe_3_O_4_-ethylene glycol nanofluid. The enhancement effect increases with the supply voltage and decreases with the Rayleigh number. They also found the natural convection heat transfer is enhanced when the volume fraction of nanoparticles is less than 0.02% but is deteriorated when the volume fraction of nanoparticles further increases. In other words, there is an optimal volume fraction of nanoparticles. In addition, combining the applied electric field and the volume fraction of nanofluid, they found that the natural convection thermal conductivity of the nanofluid with a concentration of 0.02% is the highest when the power supply voltage is 12.5 kV at a low Rayleigh number. However, once the value exceeds, the deposition of nanoparticles will have a negative impact on heat transfer, but the applied electric field can weaken the negative effect of particle deposition, which is equivalent to increasing the optimal value, thereby increasing the heat transfer rate of nanofluids.

### 2.3. Applications

While the researchers mentioned in the above section focus on the mechanism research, some other scholars are committed to application research (shown in Table 2) on enhanced heat transfer by the applied electric field. Experiment and numerical simulation are the most common methods. In experiments, the most commonly used methods to measure the thermal conductivity of nanofluids in experiments include the steady-state parallel-plate method [102], the temperature oscillation method [5] and the transient hot wire method [103], among which the transient hot wire method is the most widely employed.

Liu et al. [70] prepared Al_2_O_3_-transformer oil nanofluid with different particle sizes and concentrations by the two-step method. The thermal conductivity of the nanofluids is measured by the transient hot wire method. In addition, they also examined the enhancement ratio with different electric field strength, temperature concentration and particle size. The results showed that the electric field could enhance the thermal conductivity of nanofluids, and the enhancement effect is proportional to the concentration and temperature of nanoparticles. This study provides theoretical support for the idea of using Al_2_O_3_-transformer oil nanofluids under applied electric fields as a coolant instead of conventional transformer oil. Dhar et al. [104] also did similar research on coolant. Dhar et al. [104] also conducted similar research on coolants. They proposed and experimentally proved a new method to induce electrophoresis by applying an electric field to enhance the migration of particles from the hot zone to the cold zone, thereby improving the performance of nanofluids. They indicated that this method can considerably enhance the heat transfer of nanofluids and transiently control the thermal conductivity.

Fragelli et al. [105] have tested the possibility of nanofluids replacing tool-cutting fluids. The effect of adding silver nanoparticles with different concentrations in glycol and deionized water on the heat transfer coefficient in the presence of an electric field was studied. As the concentration of the nanofluid increases, the heat transfer coefficient first increases. When the concentration exceeds the critical value, the heat transfer coefficient decreases due to the deposition of nanoparticles on the heating surface and the increase of the viscosity of the nanofluid. However, the applied electric field can weaken the negative influence of the deposition of nanoparticles and the increase in viscosity on the heat transfer coefficient.

Heris et al. [28] summarized the previous works on two-phase closed thermosyphon (TPCP) and found that most works were on the influence of magnetic field on the performance of TPCP. Hence, they experimentally explored the influence of the applied electric field on the performance of a two-phase closed thermosyphon. In their experiment, the volume of CuO-H_2_O nanofluid accounted for 40% of the total volume of the evaporator, and the electric field was applied to the system in the voltage range of 5–20 kV. The thermal efficiency and thermal resistance of TPCP in the presence of different electric field intensities and different volume fractions were tested. It was found that using nanofluid and applied electric field at the same time can increase the thermal efficiency by up to 30% compared with pure water without an electric field.

Zhao et al. [106] explored the localized surface plasmon resonance (LSPR) effect on the latent heat of vaporization (LHV) of silver nanofluids and found that the LHV of silver nanofluids is negatively correlated with the local enhanced electric field intensity on the surface of silver nanoparticles when there is the LSPR.

Sheikholeslami et al. [107] investigated the application of electric field for augmentation of ferrofluid heat transfer in an enclosure, including double moving walls. Moreover, the influence of the shape of nanoparticles on heat transfer was also studied, and it was found that the temperature gradient of the nanofluid increases with the increase of the electric field intensity, thereby enhancing the heat transfer. They also found that the platelet shape nanoparticles can lead to the highest convection, which was also confirmed by Truong Khang et al. [95]. This may provide another theoretical support for the application of nanofluids and electric fields in biological fields.

### 2.4. Summary

In this section, the mechanism, influencing factors and influence laws of electric field on heat transfer enhancement of nanofluids are reviewed, and their applications are briefly summarized also. It is known that the applied electric field can significantly elevate the thermal conductivity of nanofluids and consequently enhance the heat transfer performance of nanofluids. Due to the mechanism of heat transfer enhancement by nanofluids has not been established, there are different views on the mechanism of heat transfer enhancement of nanofluids by an electric field. One of the reasons why electric field can enhance the heat transfer of nanofluids is it exerts an electrophoretic force, dielectrophoretic force and electrostriction force on nanofluids. Although the dominant forces are various for various nanofluids and electric fields, all can enhance the heat transfer of nanofluids. Further research on the electrophoretic force, dielectrophoretic force and electrostriction force is also a major focus in the future. Moreover, the researchers also found that Reynolds number, Rayleigh number, the volume fraction of nanoparticles, gravity, and pressure can affect the heat transfer enhancement by an applied electric field. For example, at low Reynolds number and low Rayleigh number, the heat transfer enhancement by an applied electric field is more obvious, though the mechanism needs further study. Moreover, pressure can inhibit the heat transfer of nanofluids. Gravity will destroy the electric convection generated by an electric field, thus weakening the ability of an electric field to enhance heat transfer. In practical applications, how to eliminate the influence of gravity on electric field heat transfer is another challenge.

At present, researchers are still interested in the application of nanofluids, and there is relatively little research on nanofluids in the presence of applied electric fields. However, it is found that this combination seems to be a hot topic in the research of new cooling systems on the basis of the abovementioned articles. Compared with using nanofluid as coolant only, the combined use of applied electric field and nanofluid can significantly improve the cooling effect of coolant, and the electric field can conveniently adjust the heat transfer effect of nanofluid according to the conditions so as to make the operation of cooling system more flexible. However, how to apply this combination to practice is also a problem worth considering.

## 3. Effect of Magnetic Field on Heat Transfer

### 3.1. The Mechanism of Heat Transfer Enhancement by Magnetic Field

Another hotspot of active heat transfer technology, the heat transfer enhancement by an applied magnetic field, has attracted many researchers’ attention in recent years. For the magnetic field enhanced heat transfer, some researchers proposed a theory that the major factor for the heat transfer enhancement by an applied magnetic field is the magnetic force on the magnetic fluid. The reasons could be that the applied magnetic field increases the interaction and collision between particles-particles, particles-liquid, particles-wall, and enhances the heat transfer of magnetic fluid. Wang [108] held that the apparent density of the magnetic fluid would be increased by increasing the magnetic field, while natural convection is caused by buoyancy due to the density change of the magnetic fluid. Therefore, the magnetic field can obviously promote the natural convection in the flow field by changing the apparent density of the magnetic liquid, thus enhances the heat transfer of the magnetic fluid. Shakiba et al. [109] proposed another viewpoint based on their experiments, which is the non-uniform transverse magnetic field produces Kelvin force, which produces a pair of vortices to drive the magnetic fluid to the center of the tube from both side walls. Hence, the cold boundary layer diffuses to the warm, magnetic fluid, and finally improves the heat transfer and increases the Nusselt number.

### 3.2. Effect of Magnetic Field on Heat Transfer Enhancement

#### 3.2.1. Positive Impact

Some researchers concluded that the magnetic field could enhance the heat transfer of nanofluids, and the heat transfer performance is better under higher intensity magnetic field strength. Both Mustafa et al. [110] and Shafee et al. [111] have proved such a conclusion. Mustafa et al. [110] examined the influence of magnetic field on peristaltic motion of nanofluids. The peristaltic motion can be enhanced by a powerful magnetic field and greater buoyancy caused by temperature gradient. Moreover, it is found that the temperature decreases with the increase of the magnetic field, and the cooling efficiency is significantly elevated. Shafee et al. [111] investigated the effect of Lorentz force on nanofluids between two cylinders and found the temperature gradient and heat transfer performance of nanofluid increase with the magnetic field.

Many factors, such as the direction, the inclination of the magnetic field and the distribution of the magnetic field, will considerably influence the heat transfer enhancement in the presence of a magnetic field. Wu et al. [30], Siddiqui et al. [112], Singh et al. [113], and Yousefi et al. [114] considered the influence of magnetic field direction, magnetic field inclination angle and magnetic field distribution on the heat transfer of magnetic fluid. Wu et al. [30] explored the influence of magnetic field intensity and distribution on magnetic fluid. The system diagram is shown in Figure 3. Outcomes showed that the magnetic force of magnetic fluid increases with the applied magnetic field. This can promote the natural convection of magnetic fluid, strengthen the internal energy transfer process of the magnetic fluid, consequently improve the heat transfer performance of the magnetic fluid. As for the magnetic field distribution, they found that the Nusselt number will be increased when the magnetic source becomes closer to the cold (hot) source and be instead decreased when the magnetic source is in the middle position. Siddiqui et al. [112] investigated the flow of TiO_2_-water-based nanofluid in the existence of an oblique magnetic field and found, with the increase of magnetic field intensity and angle, the fluid speed of pure water and nanofluid decreases, but the temperature increases. The Hartmann number and Nusselt number increase with the magnetic field strength. Singh et al. [113] carried out a series of parameterized studies to determine the optimal configuration and direction of the magnetic field, which can enhance convective heat transfer. The magnetic field directions are shown in Figure 4. It was found, in contrast to the Bx and By magnetic fields, the Bz magnetic field inhibits the flow fluctuation but does not inhibit convection in the enclosure. Thus, the convective heat transfer can be enhanced. It was also shown, the average Nusselt number increases with the Hartmann number. Yousefi et al. [114] simulated the influence of a non-uniform magnetic field on the forced convection heat transfer of ferrofluids. The results showed that placing two parallel wires in the flat part of the flattened tube will produce the maximum heat transfer and pressure drop of ferrofluids.

Larimi et al. [115] considered the influence of magnetic field distribution and Reynolds number on heat transfer enhancement by an applied magnetic field. Outcomes demonstrated that the applied magnetic field could increase the local Nusselt number. It also has a significant effect on the heat transfer enhancement at low Reynolds numbers like that for the applied electric field mentioned above. The effect of Reynolds number on heat transfer enhancement by the magnetic field is also considered in the study of Moghadam et al. [116]. After studying the influence of applied non-uniform magnetic field generated by current-carrying wire on the thermal physical parameters of ferrofluid, they found that the magnetic field has a great influence on heat transfer and cooling in microchannels. Moreover, the effect of the magnetic effect is remarkable at a lower Reynolds number.

Unlike other researchers who always use straight wires to induce magnetic fields, Fadaei et al. [117] utilized the finite length solenoid to induce a magnetic field and investigated the influence on the forced-convection heat transfer. The numerical results showed that the average Nusselt number increases by 30% in the presence of solenoid induced magnetic field caused by a current of 10A. They concluded that the magnetic fluid velocity would increase when the magnetic fluid moves to the porous medium by applying a magnetic field, which in turn would disturb the heat transfer boundary layer, increase the temperature gradient, thus increasing the local Nusselt number.

Yousofvand et al. [118] and Sheikholeslami et al. [119] have different views on magnetic field enhanced heat transfer. Yousofvand et al. [118] numerically analyzed magnetohydrodynamics (MHD) convection with Cu-water nanofluid. They found that when the Hartmann number is between 50 and 200, the Nusselt number increases with the Hartmann number, while the Nusselt number decreases with the Hartmann number when greater than 200. The results of Sheikholeslami et al. [119] are proved the heat transfer enhancement increases when the Hartmann number increases from 0 to 40 at *Ra* = 10^4^ and 10^5^. However, the heat transfer enhancement first increases and then decreases when the Hartmann number further increases, especially *Ra* > 10^5^. It is found there is an optimal value of magnetic field strength. When the magnetic field exceeds this critical value, it will only produce negative effects.

#### 3.2.2. Negative Effect

Interestingly, the research results of some scholars are contrary to those described in the above section. The results show that the magnetic field has no enhanced effect on the heat transfer of nanofluids and even weakens the heat transfer performance of nanofluids.

For example, Mehmood and Tabassum [120] investigated the transverse transport of Fe_3_O_4_-H_2_O nanofluid in the presence of a magnetic field. It was found that the heat transfer rate per unit area decreases with the increase of magnetic field intensity. Sheikholeslami et al. [121,122] found that the convection and the temperature gradient decrease with the magnetic field strength and Hartmann number. Alnaqi et al. [123] explored the influence of magnetic fields on the natural convection heat transfer rate of nanoparticles. The results showed that the Nusselt number increases with the Rayleigh number and decreases with the Hartmann number. They observed that the linear density of temperature near the constant temperature wall decreases with the magnetic field intensity. Thus the temperature gradient at this position decreases, so the heat transfer rate of all heat transfer mechanisms decreases. Li et al. [124] analyzed the effects of heat radiation, Eckert number and Hartmann number on the flow and heat transfer of nanofluids in the presence of Lorentz force. The results showed that the vertical velocity and Nusselt number decrease with the Hartmann number, but the temperature increases with the Hartmann number.

Wang et al. [125] investigated the coupling phenomenon of magnetohydrodynamic natural convection. Outputs revealed that the magnetic field weakens the circulation flow and heat transfer. The results of Afrand et al. [126] demonstrated that the heat transfer rate increase with the Rayleigh number and decreases with the Hartmann number. Karagiannakis et al. [127] found that the magnetic field hindered the movement of nanofluids, and thus significantly reduced the local Nusselt number owing to the inhibition of convection by magnetic fields, in which excessive nanoparticle volume fraction also can worsen heat transfer by hindering the flow of nanofluids [128]. Tran et al. [129] carried out a thermal analysis of ferrofluid under buoyancy and external force and found that the thermal plume and the average Nusselt number both decrease with the Hartmann number. Astanina et al. [130] investigated the heat transfer enhancement of natural convection of nanofluids in the presence of the external Lorentz force. According to the experimental results, the average Nusselt number and the concentration of nanoparticles are negatively correlated with the magnetic field strength.

One of the reasons why these researchers got the opposite results is that the applied magnetic field was not in the right direction, which not only weakens the heat transfer but also increased the viscosity of nanofluids, which played a negative role. Moreover, Tassone et al. [131] showed that the heat transfer of nanofluids in the existence of magnetic field increase or decrease depending on the magnetic field and the inertial strength of the fluid [113].

### 3.3. Application Research

Just as nanofluids under electric field are regarded as substitutes for new cooling systems, the application of magnetic fields and nanofluids, as shown in Table 3 in cooling systems, has attracted many researchers’ attention. Ghadiri et al. [132] examined the influence of ferrofluid on the efficiency of a PVT (photovoltaic thermal unit) system. It was found that the overall heat transfer efficiency of the PVT system could be increased by about 79% with 3 wt.% ferrofluid in the presence of a 50 Hz alternating magnetic field. Moreover, they also did an interesting comparison experiment. After studying the alternating magnetic fields, they explored the effect of a constant magnetic field on a ferrofluid and found that the thermal efficiency of the ferrofluid with a constant magnetic field was very close to that without a magnetic field.

Nanofluids are widely used in heat exchangers because of their excellent thermal conductivity. Therefore, many researchers have been exploring the influence of an applied magnetic field on the thermal conductivity of nanofluids. Chen et al. [133] used Cu-EGW (a mixture of ethylene glycol and DI-water), Al_2_O_3_-EGW and Fe_3_O_4_-EGW nanofluids to investigate the influence of magnetic field on the thermal conductivity of nanofluids. Finally, they found an interesting phenomenon, the heat transfer efficiency of Fe_3_O_4_-EGW nanofluid in the presence of the magnetic field of 100 mT is higher than that of 200 mT, but both are higher than that without a magnetic field. Their explanation is that a strong magnetic field can cause nanoparticles to deposit on the wall of the tube. This can result in the deterioration of heat transfer.

Two-phase closed thermosyphon (TPCT) is widely used in heat-recovery systems due to its high efficiency. Salehi et al. [134] hoped to enhance the heat transfer of TPCT by changing the fluid transmission characteristics and flow characteristics of the working fluid. Therefore, they used paramagnetic nanofluid as the working fluid to study the reduction of thermal resistance of thermosyphon in the existence of a magnetic field. The experimental results showed that the thermal resistance of thermosyphon decreases significantly with the concentration of nanoparticles and magnetic field strength. Accordingly, the Nusselt number increases in the presence of a magnetic field. They also investigated the effect of different concentrations of silver/water nanofluids on the heat transfer performance of TPCT under different magnetic field intensities. The results showed that TPCT performed better at the highest concentration and magnetic field strength [135].

What’s more, Shakiba et al. [109] confirmed that a non-uniform magnetic field could improve the magnetic fluid flow and enhance the cooling performance of a double pipe heat exchanger. Zhao et al. [63] found that the heat transfer and flow of nanofluids can be controlled by the magnetic and EDL. This can be a theoretical direction for the application of microfluidic devices in the existence of a magnetic field.

Different from the previous researchers, Ma et al. [136] explored the magnetohydrodynamics of ferrofluid by lattice Boltzmann method in an I-shaped multi-pipe heat exchanger. It was found that the average Nusselt number of ferrofluid decreased with the magnetic field strength, which indicated that the magnetic field had a negative effect on the heat transfer of ferrofluid.

### 3.4. Summary

In this section, the mechanism, influencing factors and influencing laws of magnetic field enhanced heat transfer of nanofluids are reviewed, and the application of a magnetic field to heat transfer of nanofluids is also briefly summarized. Different from the electric field enhanced heat transfer, the researchers’ views on magnetic field enhanced heat transfer can be divided into four categories: (a) The magnetic field can significantly enhance the heat transfer of nanofluids; (b) The magnetic field can enhance the heat transfer of nanofluids in a certain range, but it will produce negative effects when the magnetic field exceeds a certain value; (c) The magnetic field has little effect on the heat transfer of nanofluids; (d) The magnetic field weakens the heat transfer of nanofluids.

Some researchers indicated that this is caused by the different magnetic field direction and magnetic field distribution. The nanoparticles are less easy to deposit on the channel wall due to the magnetic force when the magnetic field being parallel to the main flow of nanofluid. The mass fraction of nanoparticles near the channel wall will basically remain unchanged, and the flow boundary layer will not be destroyed. However, a chain structure parallel to the flow direction will be formed, and heat transfer will be hindered. Instead, the nanoparticles will move towards the channel wall and deposit on it due to the magnetic force induced by the magnetic field perpendicular to the main flow. Moreover, more strenuous movement of nanoparticles under a stronger magnetic field will result in more destruction of the boundary layer. Moreover, particle migration increases the mass fraction of nanofluid and improves the thermal conductivity of nanofluid near the channel wall, thus reducing thermal resistance and enhancing heat transfer. In regret, the truth of this viewpoint hasn’t been verified yet.

Except for the magnetic field direction, magnetic field distribution will affect the heat transfer enhancement of nanofluids also. The results showed that the heat transfer enhancement of nanofluids by magnetic field under low Reynolds number is more significant. This is different from the conclusion of some scholars, who think that the movement of nanoparticles in a turbulent state is more intense, and the magnetic field will achieve a better strengthening effect in such a case. However, it is difficult for us to understand when the mechanism of heat transfer enhancement by an applied magnetic field has not been confirmed.

It is also found that the magnetic field has a good enhancement effect on boiling heat transfer, and they proposed that the best heat transfer performance can be obtained when the magnetic field is applied along the temperature gradient direction.

Magnetic fields have also been used to improve the heat transfer performance of nanofluids in PVT, heater, heat exchanger and TPCT, and good results have been achieved. However, Ma et al. found that the magnetic field had a negative effect on the heat transfer when the magnetic field was applied to the nanofluid in the I-tube heat exchanger. The reason may also be related to the direction of the applied magnetic field.

In a word, how to apply magnetic field correctly, especially the direction of the magnetic field, to enhance heat transfer is the most crucial problem to be considered by researchers in the future.

## 4. Conclusions

In the early stage, many scholars considered the flow of nanofluids as a single-phase flow. Actually, the interaction between nanoparticles and liquids, the movement between particles and liquids both play an important role in the convective heat transfer performance of nanofluids. At present, most researchers regard the flow of nanofluid as a two-phase flow. Current research works have not yet given a clear conclusion on the heat transfer mechanism of nanofluids and the impacting mechanism by an applied electric field and magnetic field. There are two groups of research on the enhancement of heat transfer by an electric field and magnetic field. One is to study the effect of the applied electric field and magnetic field on heat transfer characteristics and heat conduction mechanism of nanofluid. Another is for the application research of the applied electric field and magnetic field.

In this review, the related articles on electric fields and magnetic fields influencing the heat transfer of nanofluids have been reviewed for recent decades. The methods employed by researchers and the important results are presented. The purpose of this review is to explore the effect of the applied electric field or magnetic field on the heat transfer performance and mechanism of nanofluids, which can let researchers have a clear understanding of the current research status of the effect of electric or magnetic field on heat transfer of nanofluids, and to provide promising research direction for future research works for the heat transfer enhancement of nanofluids by applied electric or magnetic fields.

With the development of nanofluids in heat transfer, the application of applied electric field and magnetic field to enhance heat transfer will also have a wide prospect. However, the primary problem is to clarify the mechanism of heat transfer enhancement by applied electric fields and magnetic fields so as to control the negative effect of magnetic fields. The combination of electric field and magnetic field for heat transfer enhancement also has a great opportunity and challenge in the future.

## Figures and Tables

**Figure 1 nanomaterials-10-02386-f001:**
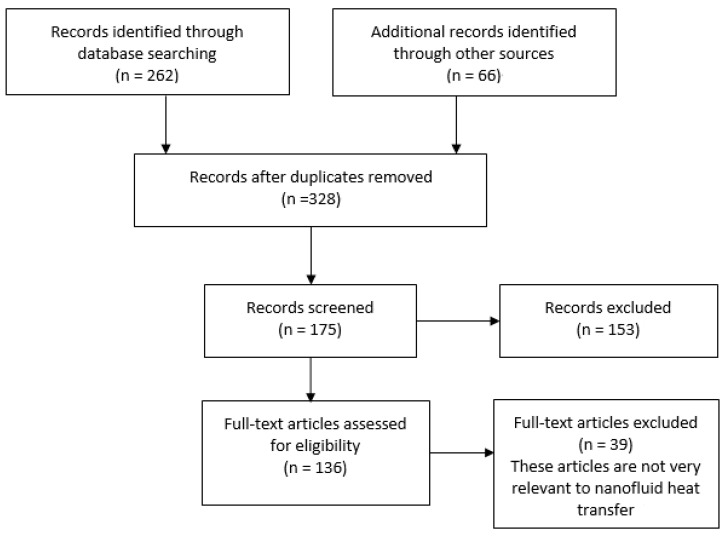
Flowchart of this survey.

**Figure 2 nanomaterials-10-02386-f002:**
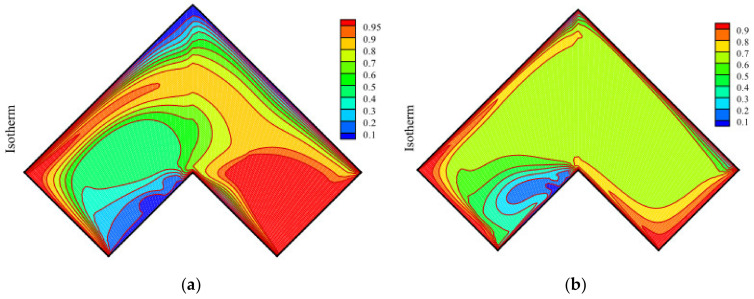
(**a**) Contours for isotherm when the applied electric field voltage is 0 kV; **(b**) Contours for isotherm when the applied electric field voltage is 10 kV, figures are reprinted from [99].

**Figure 3 nanomaterials-10-02386-f003:**
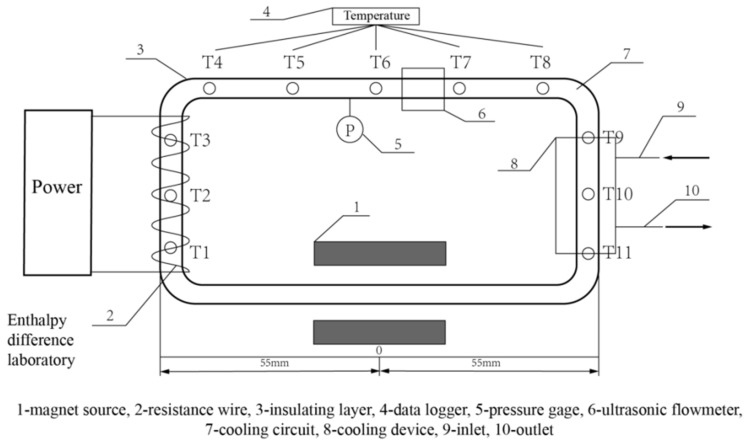
System diagram; the figure is reproduced from [30].

**Figure 4 nanomaterials-10-02386-f004:**
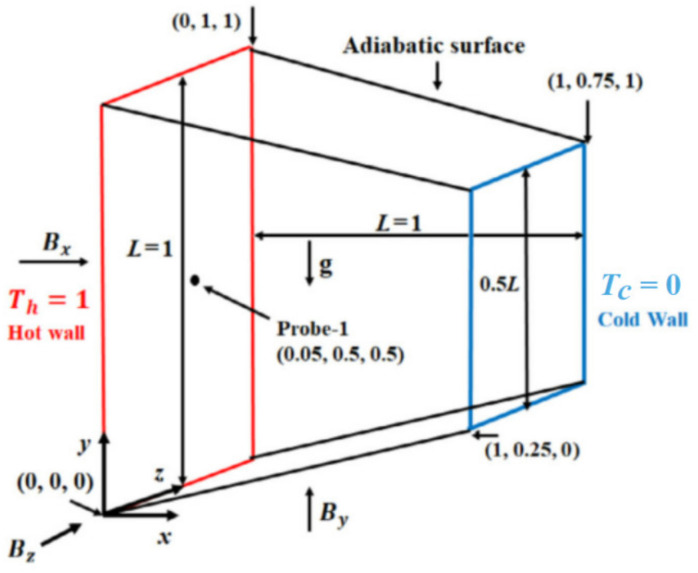
Schematic diagram of each magnetic field direction; figure is reprinted from [113], where, *T_h_* and *T_c_* and are temperatures at hot wall and cold wall, *B_x_*, *B_y_*, and *B_z_* are the magnetic strength in *x*, *y* and *z* directions, respectively.

**Table 1 nanomaterials-10-02386-t001:** Summary of influence factors for heat transfer enhancement of nanofluid by an electric field.

Author	Nanofluid	Factors	Research Methods	Main Conclusion
Bao et al. [68]	Al_2_O_3_-oil nanofluid	gravity	experimental study	The natural convection caused by gravity can counteract the electric convection generated by the electric field, thus weaken the enhancement effect of the electric field.
Saravani et al. [75]	Al_2_O_3_-H_2_O nanofluid	pressure	numerical simulation	The pressure is negatively correlated with the heat transfer of nanofluids, while the electric field is positively correlated.
Sheikholeslami et al. [76,77,78,79,80,81,82]	Fe_3_O_4_-ethylene glycol nanofluid	Reynolds number and nanoparticle volume fraction and supplied voltage	numerical simulation	Thermal conductivity increases with an increase of Reynolds number and supplied voltage. The heat transfer performance of the electric field is better at a low Reynolds number.
Sheikholeslami et al. [83]	Fe_3_O_4_-ethylene glycol nanofluid	Rayleigh number, nanoparticle volume fraction and the supplied voltage	numerical simulation	Thermal conductivity increases with the increase of Rayleigh number and supplied voltage. The heat transfer performance of the electric field is better at a low Reynolds number.
Sheikholeslami et al. [84]	Fe_3_O_4_-H_2_O nanofluid	Reynolds number, nanoparticle volume fraction and supplied voltage	numerical simulation	Thermal conductivity increases with the increase of Reynolds number and supplied voltage.
Sheikholeslami et al. [85,86,87]	Fe_3_O_4_-ethylene glycol nanofluid	radiation parameter, supplied voltage, volume fraction of nanofluid, Darcy number and Reynolds number	numerical simulation	The distortion of isotherm increases because of the rise of Darcy number, radiation parameters and Coulomb force. Thermal conductivity increases with the increase of Reynolds number and supplied voltage.
Sheikholeslami et al. [88]	Fe_3_O_4_-H_2_O nanofluid	radiation parameter, supplied voltage, volume fraction of nanofluid, Darcy number and Reynolds number	numerical simulation	The temperature gradient is positively correlated with Darcy number, radiation parameters and Coulomb force. The maximum temperature gradient is obtained when the nanoparticles are platelet shape.
Sheikholeslami et al. [89,90]	Fe_3_O_4_-ethylene glycol nanofluid	radiation parameter, supplied voltage, volume fraction of nanofluid, Darcy number and Reynolds number	numerical simulation	The heat transfer rate is the highest when the shape of nanoparticles is platelet shape. Darcy number, radiation parameter and Coulomb forces can enhance the convective heat transfer.
Sheikholeslami et al. [91]	Fe_3_O_4_-ethylene glycol nanofluid	radiation parameter, supplied voltage, volume fraction of nanofluid, Darcy number and Rayleigh number	numerical simulation	The distortion of isotherm increases because of the rise of Darcy number, radiation parameters and Coulomb force. Nusselt number increases with the argument of Darcy number, radiation parameters and Coulomb force.
Sheikholeslami et al. [92]	Fe_3_O_4_–C_2_H_6_O_2_ nanofluid	supplied voltage, permeability, radiation parameters, nanoparticles’ shape and concentration.	numerical simulation	Nusselt number increases with the argument of supplied voltage, permeability, radiation parameters and concentration of nanoparticles.
Safarnia et al. [93]	Fe_3_O_4_-H_2_O nanofluid	Reynolds number and supplied voltage	numerical simulation	Nusselt number increases with the argument of Reynolds number and supplied voltage. The heat transfer performance of the electric field is better at a low Reynolds number.
Saleem et al. [94]	Fe_3_O_4_-ethylene glycol nanofluid	supplied voltage, Darcy number, shape factor, Radiation parameter and volume fraction	numerical simulation	The distortion of isotherm increases because of the rise of Darcy number, radiation parameters and Coulomb force.
Truong Khang et al. [95]	Fe_3_O_4_-ethylene glycol nanofluid	supplied voltage, radiation parameter, nanoparticles shape factor and permeability	numerical simulation	The convection enhancement with the increase of Darcy number and supplied voltage. The heat transfer performance is positively correlated with electric field and radiation.
Lu et al. [96]	dielectric liquid	Rayleigh number, buoyancy force and mobility parameters	numerical simulation	The electric field can enhance the heat transfer more effectively for the fluid with a larger Prandtl number at a lower Rayleigh number and lower mobility parameters.
Asadzadeh et al. [33]	Fe_3_O_4_-ethylene glycol nanofluid	Rayleigh number and nanoparticle volume fraction	experimental study	The enhancement effect of the electric field increases with the rise of supply voltage and decreases with the increase of the Rayleigh number. The applied electric field can weaken the negative effect of particle deposition.

**Table 2 nanomaterials-10-02386-t002:** The application of electric field to enhance heat transfer of nanofluids.

Author	Application Direction	Research Methods	Effect
Dhar et al. [104]	Coolant	experimental study	An electric field can control the thermal conductivity transiently.
Fragelli et al. [105]	Tool cutting fluids	experimental study	An electric field can weaken the negative effect of the deposition of nanoparticles.
Heris et al. [28]	Two-phase closed thermosyphon (TPCT)	experimental study	An electric field can increase the thermal efficiency.
Zhao et al. [106]	Latent heat of vaporization (LHV)	numerical simulation	An electric field can reduce the LHV of nanofluid.

**Table 3 nanomaterials-10-02386-t003:** The application of magnetic fields to enhance heat transfer of nanofluids.

Author	Application Direction	Research Methods	Effect
Ghadiri et al. [132]	PVT	Experimental study	Increasing heat transfer efficiency by an alternating magnetic field.
Chen et al. [133]	Electric heater	Experimental study	Increasing heat transfer efficiency by an optimal magnetic field strength.
Salehi et al. [134]	TPCT	Experimental study	The magnetic field can increase the Nusselt number.
Salehi et al. [135]	TPCT	Experimental study	TPCT performed better at the magnetic field strength.
Shakiba et al. [109]	Double pipe heat exchanger	Numerical simulation	Improving the magnetic fluid flow and enhancing the cooling performance
Zhao et al. [63]	Microfluidic devices	Numerical simulation	The heat transfer and flow of nanofluid can be controlled by the magnetic and EDL.
Ma et al. [136]	Heat exchanger	Numerical simulation	Decreasing heat transfer efficiency

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
