# Peer review of "A Review on Heat Transfer of Nanofluids by Applied Electric Field or Magnetic Field"

_nanomaterials, 2020, doi:10.3390/nano10122386_

Round 1

Reviewer 1 Report

I would like to thank the authors for the opportunity of reviewing their work. In my opinion, this study contains valuable information and should be considered for publication after major revision. However, some relevant remarks can be made in relation to different aspects of the manuscript in order to improve its overall quality.

  1. Title: According to your results heat transfer can be either enhanced or aggravated by applying an electric or magnetic field. Therefore, the word “enhancement” should be changed.
  2. ABSTRACT: Totally revise this section by following these guidelines: Motivation, methodology, main results, and conclusion. Try not to repeat "applied electric field or magnetic field". Finally, highlight the innovation of the present study and correct grammatical mistakes.
  3. INTRODUCTION:
  4. Revise the introduction section. Restate the problem and write clearly your contribution. See recent reviews on this topic such as:

https://www.sciencedirect.com/science/article/abs/pii/S0370157318303302

and other relevant references.

  1. Apart for the applications regarding heat transfer, you refer to another application, namely food processing. However, there are also other important applications that should be added including strategies for:

- enhancing crop nutrition and protection. See for example: https://www.nature.com/articles/s41565-019-0439-5 and so on

- cancer therapy. See for example:

https://www.sciencedirect.com/science/article/abs/pii/S016926071930032X?via%3Dihub and other relevant references.

- wastewater decontamination. See for example:

https://www.mdpi.com/2073-4441/11/6/1135 and other relevant references.

- The subsection 2.1 (lines 98-162) should be put in the introduction section.

  1. METHODOLOGY: This section, which is a very important one, is missing in this manuscript. This is the major drawback of the present study.

Please describe how did you survey the relative literature (search engines, keywords, inclusion criteria). Also, a flowchart of the present survey methodology is required see:

http://www.prisma-statement.org/#:~:text=PRISMA%20is%20an%20evidence%2Dbased,research%2C%20particularly%20evaluations%20of%20interventions.

  1. TABLES: You can add an additional column, namely describing if the study was an experimental, numerical, or analytical.
  2. 2 and 3 sections: Change the title of these sections. For example: “Effect of electric/magnetic field on heat transfer”
  3. 3.2.2 Section: In general, natural convection tends to be influenced by Lorentz forces. In particular, the flow tends to decelerate as the Hartmann number increases. Thus, heat transfer is expected to be aggravated. This result is highlighted by several studies in the literature. This section is incomplete since a lot of studies are missing. However, a good review study must include all the relevant papers according to a well-defined methodology. For example:

https://www.sciencedirect.com/science/article/pii/S0017931018362033?via%3Dihub and other relevant references.

  1. CONCLUSIONS: State again the innovation of your study and highlight its strengths and limitations as well as suggest future direction on this field.

Finally, there is a need to improve the overall writing of the paper. Many obvious grammatical mistakes and typos were found even in the abstract (e.g in seeking for an approaches, and so on. is).

Author Response

Response to comments of reviewer 1

Thanks for the comments.

I would like to thank the authors for the opportunity of reviewing their work. In my opinion, this study contains valuable information and should be considered for publication after major revision. However, some relevant remarks can be made in relation to different aspects of the manuscript in order to improve its overall quality.

 Thank you for your constructive suggestions and opinions.

  1. Title: According to your results heat transfer can be either enhanced or aggravated by applying an electric or magnetic field. Therefore, the word “enhancement” should be changed.

Response:Revised, we deleted the word “enhancement”.

  1. ABSTRACT: Totally revise this section by following these guidelines: Motivation, methodology, main results, and conclusion. Try not to repeat "applied electric field or magnetic field". Finally, highlight the innovation of the present study and correct grammatical mistakes.

Response:We rewrote the abstract.  

  1. INTRODUCTION:
  • Revise the introduction section. Restate the problem and write clearly your contribution. See recent reviews on this topic such as:

https://www.sciencedirect.com/science/article/abs/pii/S0370157318303302

 and other relevant references.

Response:We added some sentences to highlight the problem and our main purpose.

  • Apart for the applications regarding heat transfer, you refer to another application, namely food processing. However, there are also other important applications that should be added including strategies for:

- enhancing crop nutrition and protection. See for example: https://www.nature.com/articles/s41565-019-0439-5  and so on

- cancer therapy. See for example:

https://www.sciencedirect.com/science/article/abs/pii/S016926071930032X?via%3Dihub  and other relevant references.

- wastewater decontamination. See for example:

https://www.mdpi.com/2073-4441/11/6/1135  and other relevant references.

Response:We add these applications to the paper.

  • The subsection 2.1 (lines 98-162) should be put in the introduction section.

       Response:We move these sentences to the introduction section.

  1. METHODOLOGY: This section, which is a very important one, is missing in this manuscript. This is the major drawback of the present study.

Please describe how did you survey the relative literature (search engines, keywords, inclusion criteria). Also, a flowchart of the present survey methodology is required see:

http://www.prisma-statement.org/#:~:text=PRISMA%20is%20an%20evidence%2Dbased,research%2C%20particularly%20evaluations%20of%20interventions.

 Response:We added the flowchart of this survey.

  1. TABLES: You can add an additional column, namely describing if the study was an experimental, numerical, or analytical.

Response:Revised, we added an additional column “Method”.

  1. 2 and 3 sections: Change the title of these sections. For example: “Effect of electric/magnetic field on heat transfer”

Response:We changed the title as requirement.

  1. 2.2 Section: In general, natural convection tends to be influenced by Lorentz forces. In particular, the flow tends to decelerate as the Hartmann number increases. Thus, heat transfer is expected to be aggravated. This result is highlighted by several studies in the literature. This section is incomplete since a lot of studies are missing. However, a good review study must include all the relevant papers according to a well-defined methodology. For example:

https://www.sciencedirect.com/science/article/pii/S0017931018362033?via%3Dihub  and other relevant references.

 Response: We added some of the contents mentioned above.

  1. CONCLUSIONS: State again the innovation of your study and highlight its strengths and limitations as well as suggest future direction on this field.

 Response:We stated our purpose again.

  1. Finally, there is a need to improve the overall writing of the paper. Many obvious grammatical mistakes and typos were found even in the abstract (e.g in seeking for an approaches, and so on. is).

Response:We carefully checked and revised the full text.

Reviewer 2 Report

Dear Editor,

In my opinion, the Review is an extensive and heterogenous list of different works, in most of the cases insufficiently described, all of them connected to the effect that the application of electric or magnetic fields has on the enhancement of the heat transfer. The use of the English language has a large room for improvement. For that reason, it is hard for the reader to follow the message and arrive to conclusions, even when the authors have tried to summarize the general ideas in the field, to shed light on the mechanisms that are behind the effects of the fields.

The authors should devote a major effort to describe:

  • The role of the electrophoretic, dielectric electrophoretic and electrostrictive forces. It is discussed throughout all the manuscript, so I think that the authors should devote at least some lines to describe these forces and the differences between them.
  • What is the morphology and dynamics of the structures induced by the external fields, and how are these structures connected to the desired improvement?
  • How the heat radiation improves the heat transfer performance of nanofluids
  • Since they are central parameters, they should define, at least, the Hartmann, Darcy, Rayleigh, Nusselt and Prandtl numbers.
  • In the summary Section 3.4, the effect of the magnetic field on the nanofluids is mainly described in terms of the interaction of the nanoparticles with the tube walls, which seems to be quite system-specific.

Minor comments:

I miss more references in the first paragraph.

Page 2, Lines 55, 59 and 67. Please, describe the nanofluids used by Rao et al., Jafary et al. and Wang et al.

You repeat four or five times throughout the manuscript that the enhancement in the heat transfer of nanofluids due to the application of electric or magnetic fields has attracted the attention of many researchers. (Page 1, lines 11 and 30, Page 2 lines 43 and 92) You should not be so repetitive.

Lines 110, 126, 148, 177, 182, 185,333, 401... In many times, you say that some authors “believe” in certain facts or criteria. I would try to avoid this expression. You should explain the facts that push these authors to opt for those ideas.

Non-described acronyms: MD (page 3, line 112) EHD (page 5, line 205) MHD (page 7, line 216) CVFEM (page 7, line 224) TPCT and LHV in table 2

Page 3, line 118: How the aggregation of nanoparticles can alter on the evaporation of the fluid?

Page 3, line 119; “Proposed the Dh theorem, where h is related to the volume fraction of the nanoparticles”. From reading this sentence, the reader will not understand what the Dh theorem is, and how is it connected to the volume fraction of the nanoparticles.

Line 127 “properties of nanofluids within the boundary layer”. What boundary layer is this?

Line 145 “a gradual decrease in potential” What potential are you referring?

Line 149. The surface charge and the electric layer are not mechanisms.

Line 155 “in the presence of interface layer, aggregation…” What does it mean?

Line 160 “vibration state of Ar atoms”. You should describe first the system used by Mitiche et al. in reference 61. On the other hand, you say that “the vibration frequency of nanoparticles is consistent with that”. What do you mean?

Line 179 “zeta potential on the surface of the nanoparticles”. The zeta potential is not defined on the surface of the nanoparticles.

Line 180 “enhance the orbit of nanoparticles”. Which is this orbit?

Line 181 “in a large range of positions”. What does it mean?

Lines 182, 186 “the enhancement of the electric field”. What enhancement are you talking about?

Line 184 “destroy the boundary layer”. Which is this boundary layer?

Line 186 “the viscosity of nanofluids is small at a certain electric field intensity”. Why?

Line 189 “In addition”. I would not start a new paragraph with this expression.

Line 190 “primary electric adhesion”. “secondary electric adhesion”. What is this? I appreciate that the authors introduce some terms to the non-specialized reader.

Line 201 “generate heat flow ahead of temperature gradient”. What does it mean?

Line 209 “the heat transfer laws of nanofluids in an applied electric field”. Please, try to express better these ideas.

Table 1 is quite repetitive (for example, rows 3 and 4 are identical). I think that the ideas here summarized should be more extensively developed in a new paragraph.

215 “the heat transfer law of nanofluid”. Which is this law?

219 “the electric field increases the velocity of nanofluids”. What do you mean with this “velocity of nanofluids”?

Line 225 “without induced magnetic field”. I thought that in this section you were describing the effect of the electric field.

Line 228 “will make the isotherm more complex”. Which isotherm are you referring in this sentence?

Figure 1. What is the information that you want to supply with this Figure 1? On the other hand, in line 230 you say, “when other parameters are fixed”. Which are the other parameters? Could you describe the system to whom the Figure corresponds? Besides, you must rewrite the caption. In the actual version you repeat twice that the Figure is reprinted from [73].

Line 263. “The effect of electric field on heat transfer is obvious in the inexistence of gravity, which increases with the enhancement of voltage and heat flux”. The sentence is difficult to understand.

Line 287 “the isotherms near the wall will become denser”. What do you mean? How can be dense an isotherm?

Line 363, Section 2.4. This section is very important, and here there is a large room for improvement.

Line 400 “Shakiba et al. [106] proposed another viewpoint based on their experiments. They believed that the non-uniform transverse magnetic field produces Kelvin force which produces a pair of vortices to drive the magnetic fluid to the centre of the tube from both side walls.” The sentence is quite unintelligible. In any case, the reasons here given seem to be quite specific of the system used in the experiments. Hence, the latter should be described.

Etc…

On the other hand, I insist that there is substantial room for improvement in the use of the English language. I include only a few examples:

Page 1:

Line 11 “have been devoting in seeking for an approaches”

Line 14 “an overview involving the mechanism”

Please, avoid using the expression “and so on”. (Lines 16, 28, 32, 280…)

Line 16 “The heat transfer enhancement of nanofluid by applied electric field…”. In fact, the last phrase in the abstract is very large and confusing. It basically says that you present the mechanism involved in the enhancement of the heat transfer after applying an external field to clarify the mechanism of heat and mass transfer enhancement, which is redundant.

Line 23 “temperature gradient can transport”

Line 33 “While, ”

Line 38: “Nanofluid is a new type of heat transfer medium which is uniform, stable and high thermal conductivity by dispersing metal or non-metallic nanoparticles into translational media such as water, alcohol and oil”

Line 53: “and the thermal conductivity…”

Line 94: “review on the heat transfer enhancement by nanofluids in the presence of applied electric field and magnetic field in the past decade”

Line 102 “It is confirmed by many published results”

Line 107 “make the base fluid into a suspension”

Line 121 “aggregation morphology of nanoparticles”

Line 138 “the heat transfer of nanofluids may exist non-Fourier conduction”

Line 146 The whole sentence is not understandable.

Line 164 “such as”

Line 177 “the fluid is covered by electric field”

Line 195 “charged nanoparticle double layers”

Line 197 “the calculation method of direct energy transfer between nanoparticle”

Line 222 “will increase as the values of permeability”

Line 234 “one thing is the same, in which nanofluids are subjected to the external force”.

Line 246 “when the dominant force is different (different to what?), the electric Rayleigh number is proportion to them is also different”.

Line 250 “whether the electric field is uniform influences the force caused by electric field greatly, in which nanofluids under uniform electric field are mainly affected by electrophoretic force, while nanofluids under non-uniform electric field are mainly affected by electrostriction force. This is different from the single-phase flow.”

Line 262: “enhanced by electric field”. You use very often this linguistic structure, that I think is not correct. Instead, try “enhanced via the application of” “through the application” “by applying”…

Line 262 “by electric field, outcomes”. I think that here a period is more appropriate.

Line 269: “heat transfer, they found”. Again, I think that a period is more appropriate.

Line 276: “It is found that, the pressure is…” Here, the comma is useless.

Line 281: “the effect of applied field”. This linguistic structure is repeated throughout the manuscript. I think that it would be better “the effect of the applied field”.

Line 293: “concluded a similar conclusion”

Line 305: “nanoparticle less than”

Line 306: “In other word”

Line 307: “In addition, combined with the applied electric field and the nanofluid volume fraction,”

Line 310: The last sentence of the paragraph is too long, and unintelligible.

Line 330: The last sentence of the page is too long, and unintelligible.

Line 336. The second sentence in the paragraph is too long, and unintelligible.

Line 341. The last sentence of the paragraph is unintelligible.

Line 355. “in the presence of LSPR”

Line 359 “temperature gradient of nanofluids”

Line 368. This sentence could be improved.

Line 366. “It is known from the published results, the applied electric field”

Line 363, Section 2.4. This section is very important, and here there is a large room for improvement.

Line 395: “the underlying mechanism for the heat transfer enhancement by applied magnetic field is the magnetic force on magnetic fluid.” A force is not a mechanism. Try to improve the structure of the sentences.

Line 396: “The reasons could be that, the”

Line 398: “This theory is similar to that by applied electric field.”

Etc…

Author Response

Response to reviewer 3

In my opinion, the Review is an extensive and heterogenous list of different works, in most of the cases insufficiently described, all of them connected to the effect that the application of electric or magnetic fields has on the enhancement of the heat transfer. The use of the English language has a large room for improvement. For that reason, it is hard for the reader to follow the message and arrive to conclusions, even when the authors have tried to summarize the general ideas in the field, to shed light on the mechanisms that are behind the effects of the fields.

The authors should devote a major effort to describe:

  1. The role of the electrophoretic, dielectric electrophoretic and electrostrictive forces. It is discussed throughout all the manuscript, so I think that the authors should devote at least some lines to describe these forces and the differences between them.

Response:Revised, we add the introduction of electrophoretic, dielectric electrophoretic and electrostrictive forces.

  1. What is the morphology and dynamics of the structures induced by the external fields, and how are these structures connected to the desired improvement?

Response:The external field produces an electric convection in the flow field, which makes the fluid move more violently and thus enhances the heat transfer of the fluid. The magnetic field can obviously promote the natural convection in the flow field by changing the apparent density of the magnetic liquid, thus enhances the heat transfer of the magnetic fluid. We added this in line 286 and 520 respectively.

  1. How the heat radiation improves the heat transfer performance of nanofluids

Response:The heat radiation can increase the temperature gradient of the nanofluid and thus improves the heat transfer performance of the nanofluid. We add this sentence in the paper.

  1. Since they are central parameters, they should define, at least, the Hartmann, Darcy, Rayleigh, Nusselt and Prandtl numbers.

Response: We explained what these parameters do at the end of introduction section.

  1. In the summary Section 3.4, the effect of the magnetic field on the nanofluids is mainly described in terms of the interaction of the nanoparticles with the tube walls, which seems to be quite system-specific.

Response: It's our mistake that did not describe it clearly, the “tube walls” in this paper we want to explain include many channel walls such as tube walls and cavity walls, and we changed the statement.

Minor comments:

I miss more references in the first paragraph.

Page 2, Lines 55, 59 and 67. Please, describe the nanofluids used by Rao et al., Jafary et al. and Wang et al.

Response: Revised, we add the nanofluids used by Rao et al., Jafary et al. and Wang et al.

You repeat four or five times throughout the manuscript that the enhancement in the heat transfer of nanofluids due to the application of electric or magnetic fields has attracted the attention of many researchers. (Page 1, lines 11 and 30, Page 2 lines 43 and 92) You should not be so repetitive.

Response: We revised the sentences mentioned above.

Lines 110, 126, 148, 177, 182, 185,333, 401... In many times, you say that some authors “believe” in certain facts or criteria

Response: We revised the statement.

Reviewer 3 Report

The paper deals with heat transfer enhancement of nanofluids and how this can be influenced further by electric or magnetic fields. The paper is well structured, in first mentioning the mechanisms, then the applications and then an intermediary summary. However, some issues should be discussed.

  • Lines 32-34: the phrase is not so well written, it should be rewritten
  • In general the English use should be checked as well as typos
  • The authors state in lines 98-103 the different mechanisms responsible for heat transfer in nanofluids. They omit one important mechanisms that takes place in solids at nanoscale (which is the case within the nanoparticles themselves), i.e. local effects. The authors should discuss this, since it can reduce the enhancement and could be quite hindering. Information on this can be found in “International Journal of Nanoscience, 2014, 13(3), 1450022”. Actually, the authors make some hints to this in lines 136-139. Maybe this discussion should be elaborated a bit in these lines.
  • What is really missing is a clear summary at the end of the paper regrouping everything written in the paper, maybe in the form of a diagram or structured figure or the like, where the reader might be instructed what to do if a certain outcome of an experiment is wanted.
  • Also, some physics should be added to this last diagram. The authors speak about laws (lines 209, 216, 237, 364 and 544) but there isn’t any in the paper. It might not be the purpose, and such thing is not necessary to do through the whole paper, but at the end the reader might want to know what physical laws are of importance, if one would like to perform modelling

Author Response

Response to reviewer 2

The paper deals with heat transfer enhancement of nanofluids and how this can be influenced further by electric or magnetic fields. The paper is well structured, in first mentioning the mechanisms, then the applications and then an intermediary summary. However, some issues should be discussed.

  • Lines 32-34: the phrase is not so well written, it should be rewritten

Response:We revised according to the comments.

  • In general the English use should be checked as well as typos

Response:We carefully checked and revised the full text.

  • The authors state in lines 98-103 the different mechanisms responsible for heat transfer in nanofluids. They omit one important mechanisms that takes place in solids at nanoscale (which is the case within the nanoparticles themselves), i.e. local effects. The authors should discuss this, since it can reduce the enhancement and could be quite hindering. Information on this can be found in “International Journal of Nanoscience, 2014, 13(3), 1450022”. Actually, the authors make some hints to this in lines 136-139. Maybe this discussion should be elaborated a bit in these lines.

Response:Revised, we add relative content in revised manuscript.

  • What is really missing is a clear summary at the end of the paper regrouping everything written in the paper, maybe in the form of a diagram or structured figure or the like, where the reader might be instructed what to do if a certain outcome of an experimentis wanted.

Response:Revised, we rewrote the conclusion according to the comment of reviewer.

  • Also, some physics should be added to this last diagram. The authors speak about laws (lines 209, 216, 237, 364 and 544) but there isn’t any in the paper. It might not be the purpose, and such thing is not necessary to do through the whole paper, but at the end the reader might want to know what physical laws are of importance, if one would like to perform modelling

Response:Revised, we add the law in line 309.

Round 2

Reviewer 1 Report

It can be accepted in its present form.

Author Response

Thanks!

Reviewer 2 Report

I regret to maintain my previous decision. I still consider that the work is not properly organized, and that the use of the English language has a large room for improvement. I consider that the manuscript does not reach the level demanded in Nanomaterials. So, I recommend the authors to devote an extra effort to write a better structured and more understandable text.

Author Response

Sorry, it’s our fault for missing some minor comments for review report (round 1) of reviewer 2. Our modifications are as follows:

Non-described acronyms: MD (page 3, line 112) EHD (page 5, line 205) MHD (page 7, line 216) CVFEM (page 7, line 224) TPCT and LHV in table 2

Response: We explained these acronyms in line 113, 248, 610, 267 and table 2.

Page 3, line 118: How the aggregation of nanoparticles can alter on the evaporation of the fluid?

Response: The aggregation of nanoparticles can build a low thermal resistance channel, which can reduce the loss of heat transfer of nanofluids.

Page 3, line 119; “Proposed the Dh theorem, where h is related to the volume fraction of the nanoparticles”. From reading this sentence, the reader will not understand what the Dh theorem is, and how is it connected to the volume fraction of the nanoparticles.

Response: Dh theorem indicates that the evaporation time of the liquid is proportional to the h power of the initial diameter of the droplet. Besides, h is related to the volume fraction and distribution of nanoparticles, with the increase of the volume fraction of lyophobic nanoparticles, h will be greater, while h will be smaller with the increase of volume fraction of lyophilic nanoparticles. We explained Dh theorem and the correlation between h and the volume fraction of nanofluids in line 121-125.

Line 127 “properties of nanofluids within the boundary layer”. What boundary layer is this?

Response: It is flow boundary layer. We added this in line 134.

Line 145 “a gradual decrease in potential” What potential are you referring?

Response: It is electric potential. We revised it in line 151.

Line 149. The surface charge and the electric layer are not mechanisms.

Response: We revised it in line 155. 

Line 155 “in the presence of interface layer, aggregation…” What does it mean?

Response: The meaning we want to express is the researcher compared the effects of interface layer, aggregate and electric double layer on the effective thermal conductivity of TiO2-water nanofluids. We revised it in line 160-162.

Line 160 “vibration state of Ar atoms”. You should describe first the system used by Mitiche et al. in reference 61. On the other hand, you say that “the vibration frequency of nanoparticles is consistent with that”. What do you mean?

Response: We added the system and revised the sentence mentioned above in line 166-171.

Line 179 “zeta potential on the surface of the nanoparticles”. The zeta potential is not defined on the surface of the nanoparticles.

Response: We added the definition of zeta potential in line 215-217.

Line 180 “enhance the orbit of nanoparticles”. Which is this orbit?

Response: It means the range of movement of nanoparticles in the base liquid. We revised it in line 217.

Line 181 “in a large range of positions”. What does it mean?

Response: It means that nanoparticles can transfer energy in a wider range. We revised it in line 218.

Lines 182, 186 “the enhancement of the electric field”. What enhancement are you talking about?

Response: It's our responsibility that did not to express it clear. It means the enhancement of heat transfer of nanofluids by applied electric field. We revised it in line 219 and 223.

Line 184 “destroy the boundary layer”. Which is this boundary layer?

Response: It’s flow boundary layer. We added it in line 222.

Line 186 “the viscosity of nanofluids is small at a certain electric field intensity”. Why?

Response: It’s our mistake that had a wrong statement. The acceleration effect of the movement of nanoparticles by electric field is better when viscosity is low, and the increase of temperature can reduce the viscosity of nanofluids. We revised it in line 225-229.

Line 189 “In addition”. I would not start a new paragraph with this expression.

Response: We used moreover to replace it in line 230.

Line 190 “primary electric adhesion”. “secondary electric adhesion”. What is this? I appreciate that the authors introduce some terms to the non-specialized reader.

Response: Electric adhesion is an electrostatic effect in which two surfaces shrink in the existence of electric field. We explained it in line 234.

Line 201 “generate heat flow ahead of temperature gradient”. What does it mean?

Response: It means that the direct energy exchange between nanoparticles may generate heat flow before the temperature gradient produces heat flow. We revised it in line 243.

Line 209 “the heat transfer laws of nanofluids in an applied electric field”. Please, try to express better these ideas.

Response: We revised it in 253.

Table 1 is quite repetitive (for example, rows 3 and 4 are identical). I think that the ideas here summarized should be more extensively developed in a new paragraph.

Response: Due to many articles we read relate to the same factors for heat transfer enhancement of nanofluid by electric field, the conclusions in Table 1 look very similar. Table 1 is a summary of what we have done below, where we have expanded on these factors.

215 “the heat transfer law of nanofluid”. Which is this law?

Response: We added the law in line 260-261.

219 “the electric field increases the velocity of nanofluids”. What do you mean with this “velocity of nanofluids”?

Response: We used flow rate to replace velocity in line 264.

Line 225 “without induced magnetic field”. I thought that in this section you were describing the effect of the electric field.

Response: We deleted this sentence.

Line 228 “will make the isotherm more complex”. Which isotherm are you referring in this sentence?

Response: It is the isotherm near the tank wall, we revised it in line 272.

Figure 1. What is the information that you want to supply with this Figure 1? On the other hand, in line 230 you say, “when other parameters are fixed”. Which are the other parameters? Could you describe the system to whom the Figure corresponds? Besides, you must rewrite the caption. In the actual version you repeat twice that the Figure is reprinted from [73].

Response: We want to show the variation of isotherm distribution near the wall under different voltages. Other parameters are Reynolds number, Darcy number, radiation parameter and heat flux. We added these parameters in line 275 and revised the caption of figure 1.

Line 263. “The effect of electric field on heat transfer is obvious in the inexistence of gravity, which increases with the enhancement of voltage and heat flux”. The sentence is difficult to understand.

Response: It means that the effect of electric field on heat transfer is obvious when there is no gravity, and the effect increases with the enhancement of voltage and heat flux. We revised it in line 309.

Line 287 “the isotherms near the wall will become denser”. What do you mean? How can be dense an isotherm?

Response: We want to express that the distribution of isotherms near the wall will become denser and we changed our expression in line 332.

Line 363, Section 2.4. This section is very important, and here there is a large room for improvement.

Response: We added some content in this section.

Line 400 “Shakiba et al. [106] proposed another viewpoint based on their experiments. They believed that the non-uniform transverse magnetic field produces Kelvin force which produces a pair of vortices to drive the magnetic fluid to the centre of the tube from both side walls.” The sentence is quite unintelligible. In any case, the reasons here given seem to be quite specific of the system used in the experiments. Hence, the latter should be described.

Response: We indicated that these researchers put forward this theory based on their experiments in this section, and due to the horizontal pipeline used in this research is universal, we also put their theory in section 1.

On the other hand, I insist that there is substantial room for improvement in the use of the English language. I include only a few examples:

Page 1:

Line 11 “have been devoting in seeking for an approaches”

Line 14 “an overview involving the mechanism”

Please, avoid using the expression “and so on”. (Lines 16, 28, 32, 280…)

Line 16 “The heat transfer enhancement of nanofluid by applied electric field…”. In fact, the last phrase in the abstract is very large and confusing. It basically says that you present the mechanism involved in the enhancement of the heat transfer after applying an external field to clarify the mechanism of heat and mass transfer enhancement, which is redundant.

Line 23 “temperature gradient can transport”

Line 33 “While, ”

Line 38: “Nanofluid is a new type of heat transfer medium which is uniform, stable and high thermal conductivity by dispersing metal or non-metallic nanoparticles into translational media such as water, alcohol and oil”

Line 53: “and the thermal conductivity…”

Line 94: “review on the heat transfer enhancement by nanofluids in the presence of applied electric field and magnetic field in the past decade”

Line 102 “It is confirmed by many published results”

Line 107 “make the base fluid into a suspension”

Line 121 “aggregation morphology of nanoparticles”

Line 138 “the heat transfer of nanofluids may exist non-Fourier conduction”

Line 146 The whole sentence is not understandable.

Line 164 “such as”

Line 177 “the fluid is covered by electric field”

Line 195 “charged nanoparticle double layers”

Line 197 “the calculation method of direct energy transfer between nanoparticle”

Line 222 “will increase as the values of permeability”

Line 234 “one thing is the same, in which nanofluids are subjected to the external force”.

Line 246 “when the dominant force is different (different to what?), the electric Rayleigh number is proportion to them is also different”.

Line 250 “whether the electric field is uniform influences the force caused by electric field greatly, in which nanofluids under uniform electric field are mainly affected by electrophoretic force, while nanofluids under non-uniform electric field are mainly affected by electrostriction force. This is different from the single-phase flow.”

Line 262: “enhanced by electric field”. You use very often this linguistic structure, that I think is not correct. Instead, try “enhanced via the application of” “through the application” “by applying”…

Line 262 “by electric field, outcomes”. I think that here a period is more appropriate.

Line 269: “heat transfer, they found”. Again, I think that a period is more appropriate.

Line 276: “It is found that, the pressure is…” Here, the comma is useless.

Line 281: “the effect of applied field”. This linguistic structure is repeated throughout the manuscript. I think that it would be better “the effect of the applied field”.

Line 293: “concluded a similar conclusion”

Line 305: “nanoparticle less than”

Line 306: “In other word”

Line 307: “In addition, combined with the applied electric field and the nanofluid volume fraction,”

Line 310: The last sentence of the paragraph is too long, and unintelligible.

Line 330: The last sentence of the page is too long, and unintelligible.

Line 336. The second sentence in the paragraph is too long, and unintelligible.

Line 341. The last sentence of the paragraph is unintelligible.

Line 355. “in the presence of LSPR”

Line 359 “temperature gradient of nanofluids”

Line 368. This sentence could be improved.

Line 366. “It is known from the published results, the applied electric field”

Line 363, Section 2.4. This section is very important, and here there is a large room for improvement.

Line 395: “the underlying mechanism for the heat transfer enhancement by applied magnetic field is the magnetic force on magnetic fluid.” A force is not a mechanism. Try to improve the structure of the sentences.

Line 396: “The reasons could be that, the”

Line 398: “This theory is similar to that by applied electric field.”

Response: Thanks for these minor comments about the use of the English language, we revised our paper according to the minor comments and highlighted the modifications in yellow.

Reviewer 3 Report

The authors have considerably improved this version of the paper. A final check of the English use is recommended.

Author Response

Thanks!